# Sparse Attentive Backtracking:
# Temporal Credit Assignment Through Reminding

**Nan Rosemary Ke**[1,2], **Anirudh Goyal**[1], **Olexa Bilaniuk**[1], **Jonathan Binas**[1],
**Michael C. Mozer**[3], **Chris Pal**[1,2,4], **Yoshua Bengio**[1†]
[1] Mila, Université de Montréal
[2] Mila, Polytechnique Montréal
[3] University of Colorado, Boulder
[4] Element AI
[†]CIFAR Senior Fellow.

## Abstract

Learning long-term dependencies in extended temporal sequences requires credit assignment to events far back in the past. The most common method for training recurrent neural networks, back-propagation through time (BPTT), requires credit information to be propagated backwards through every single step of the forward computation, potentially over thousands or millions of time steps. This becomes computationally expensive or even infeasible when used with long sequences. Importantly, biological brains are unlikely to perform such detailed reverse replay over very long sequences of internal states (consider days, months, or years.) However, humans are often reminded of past memories or mental states which are associated with the current mental state. We consider the hypothesis that such memory associations between past and present could be used for credit assignment through arbitrarily long sequences, propagating the credit assigned to the current state to the associated past state. Based on this principle, we study a novel algorithm which only back-propagates through a few of these temporal skip connections, realized by a learned attention mechanism that associates current states with relevant past states. We demonstrate in experiments that our method matches or outperforms regular BPTT and truncated BPTT in tasks involving particularly long-term dependencies, but without requiring the biologically implausible backward replay through the whole history of states. Additionally, we demonstrate that the proposed method transfers to longer sequences significantly better than LSTMs trained with BPTT and LSTMs trained with full self-attention.

## 1   Introduction

Humans have a remarkable ability to *remember* events from the distant past which are associated with the current mental state (Ciaramelli et al., 2008). Most experimental and theoretical analyses of memory have focused on understanding the deliberate route to memory formation and recall. But automatic reminding—when memories pop into one's head—can have a potent influence on cognition. Reminding is normally triggered by contextual features present at the moment of retrieval which match distinctive features of the memory being recalled (Berntsen et al., 2013; Wharton et al., 1996), and can occur more often following unexpected events (Read & Ian, 1991). Thus, an individual's current state of understanding can trigger reminding of a past state. Reminding can provide distracting sources of irrelevant information (Forbus et al., 1995; Novick, 1988), but it can also serve a useful computational role in ongoing cognition by providing information essential to decision making (Benjamin & Ross, 2010).

In this paper, we identify another possible role of reminding: to perform credit assignment across long time spans. Consider the following scenario. As you drive down the highway, you hear an unusual popping sound. You think nothing of it until you stop for gas and realize that one of your tires has deflated, at which point you are suddenly reminded of the pop. The reminding event helps determine the cause of your flat tire, and probably leads to synaptic changes by which a future pop sound while driving would be processed differently. Credit assignment is critical in machine learning. Back-propagation is fundamentally performing credit assignment. Although some progress has been made toward credit-assignment mechanisms that are functionally equivalent to back-propagation (Lee et al., 2014; Scellier & Bengio, 2016; Whittington & Bogacz, 2017), it remains very unclear how the equivalent of back-propagation through time, used to train recurrent neural networks (RNNs), could be implemented by brains. Here we explore the hypothesis that an associative reminding process could play an important role in propagating credit across long time spans, also known as the problem of learning long-term dependencies in RNNs, i.e., of learning to exploit statistical dependencies between events and variables which occur temporally far from each other.

## 1.1 Credit Assignment in Recurrent Neural Networks

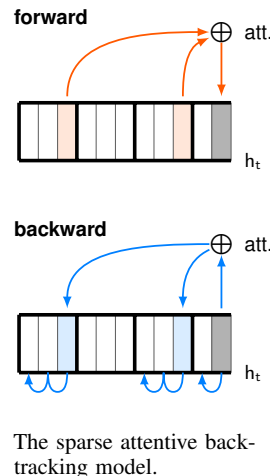

The sparse attentive back-tracking model.

RNNs are used to processes sequences of variable length. They have achieved state-of-the-art results for many machine learning sequence processing tasks. Examples where models based on RNNs shine include speech recognition (Miao et al., 2015; Chan et al., 2016), image captioning (Vinyals et al., 2015; Lu et al., 2017), machine translation (Luong et al., 2015).

It is common practice to train RNNs using gradients computed with *back-propagation through time* (BPTT), wherein the network states are unrolled in time over the whole trajectory of discrete time steps and gradients are back-propagated through the unrolled graph. The network unfolding procedure of BPTT does not seem biologically plausible. It requires storing and playing back these events (in reverse order) using the same recurrent weights to combine error signals with activities and derivatives at previous time points. The replay is initiated only at the end of a trajectory of $T$ time steps, and thus requires memorization of a large number of states. If a discrete time instant corresponds to a saccade (about 200-300ms,) then a trajectory of 100 days would require replaying back computations through over 42 million time steps. This is not only inconvenient, but more importantly a small error to any one of these events could either vanish or blow up and cause catastrophic outcomes. Also, if this unfolding and back-propagation is done only over shorter sequences, then learning typically will not capture longer-term dependencies linking events across larger temporal spans then the length of the back-propagated trajectory.

What are the alternatives to BPTT? One approach we explore here exploits *associative reminding* of past events which may be triggered by the current state and added to it, thus making it possible to propagate gradients with respect to the current state into approximate gradients in the state corresponding to the recalled event. The approximation comes from not backpropagating through the unfolded ordinary recurrence across long time spans, but only through this memory retrieval mechanism. Completely different approaches are possible but are not currently close to BPTT in terms of learning performance on large networks, such as methods based on the online estimation of gradients (Ollivier et al., 2015). Assuming that no exact gradient estimation method is possible (which seems likely) it could well be that brains combine multiple estimators.

In machine learning, the most common practical alternative to full BPTT is *truncated BPTT* (TBPTT) Williams & Peng (1990). In TBPTT, a long sequence is sliced into a number of (possibly overlapping) subsequences, gradients are backpropagated only for a fixed, limited number of time steps into the past, and the parameters are updated after each backpropagation through a subsequence. Unfortunately, this truncation makes capturing dependencies across distant timesteps nigh-impossible, because no error signal reaches further back into the past than TBPTT's *truncation length*.

Neurophysiological findings support the existence of remembering memories and their involvement in credit assignment and learning in biological systems. In particular, hippocampal recordings in rats indicate that brief sequences of prior experience are replayed both in the awake resting state and during sleep, both of which conditions are linked to memory consolidation and learning (Foster & Wilson, 2006; Davidson et al., 2009; Gupta et al., 2010; Ambrose et al., 2016). Thus, the mental

look back into the past seems to occur exactly when credit assignment is to be performed. Thus, it is plausible that hippocampal replay could be a way of doing temporal credit assignment (and possibly BPTT) on a short time scale, but here we argue for a solution which could handle credit assignment over much longer durations.

### 1.2 Novel Credit Assignment Mechanism: Sparse Attentive Backtracking

Inspired by the ability of brains to selectively reactivate memories of the past based on the current context, we propose here a novel solution called *Sparse Attentive Backtracking* (SAB) that incorporates a differentiable, sparse (hard) attention mechanism to select from past states. Inspired by the cognitive analogy of reminding, SAB is designed to retrieve one or very few past states. This may also be advantageous in focusing the credit assignment, although this hypothesis remains to be tested. SAB meshes well with TBPTT, yet allows gradient to propagate over distances far in excess of the TBPTT truncation length. We experimentally answer affirmatively the following questions:

**Q1. Can Sparse Attentive Backtracking (SAB) capture long-term dependencies?** SAB captures long-term dependencies. See results for 7 tasks supporting this in §4.

**Q2. Generalization and transfer ability of SAB?** See the strong transfer results in §4.

**Q3. How does SAB perform compared to the Transformers** (Vaswani et al., 2017)**?** SAB outperforms the Transformers (comparison in §4).

**Q4. Is sparsity important for SAB and does it learn to retrieve meaningful memories?** See the results on the Importance of Sparsity and Table 4 in §4.

## 2 Related Machine Learning Work

**Skip-connections and gradient flow** Neural architectures such as Residual Networks (He et al., 2016) and Dense Networks (Huang et al., 2016) allow information to skip over convolutional processing blocks of an underlying convolutional network architecture. This construction provably mitigates the vanishing gradient problem by allowing the gradient at any given layer to be bounded. Densely-connected convolutional networks alleviate the vanishing gradient problem by allowing a direct path from any layer in the network to the output layer. In contrast, in this work we propose and explore what one might regard as a form of dynamic skip connection, modulated by an attention mechanism corresponding to a reminding process, which matches the current state with an older state that is retrieved from memory.

Recurrent neural networks with skip-connections in time can allow information to flow over much longer time spans. These skip-connections can have either a fixed time span such as in hierarchical El Hihi & Bengio (1996) or clockwork Koutnik et al. (2014) RNNs, or a dynamic time span such as in Chung et al. (2016); Mozer et al. (2017); Ke et al. (2018). All of these models still need to be trained with full BPTT, which requires a full replay of past events. Designs also exist based on *wormhole connections*, implemented as differentiable reads and writes to external memories, as in Gulcehre et al. (2017). Also, as noted in Kádár et al. (2018), with highly complex architectures, training procedure and implementations might hinder their utility.

**The transformer network** The *Transformer* network (Vaswani et al., 2017) takes sequence processing using attention to its logical extreme – using attention *only*, not relying on RNNs at all. The attention mechanism is a softmax not over the sequence itself but over the outputs of the previous self-attention layer. In order to attend to multiple parts of the layer outputs simultaneously, the Transformer uses 8 small attention "heads" per layer (instead of a single large head) and combines the attention heads' outputs by concatenation. No attempt is made to make the attention weights sparse, and the authors do not test their models on sequences of length greater than the intermediate representations of the Transformer model. With brains clearly involving a recurrent computation, this approach would seem to miss an important characteristic of biological credit assignment through time. Another implausible aspect of the Transformer architecture is the simultaneous access to (and linear combination of) all past memories (as opposed to a handful with SAB.)

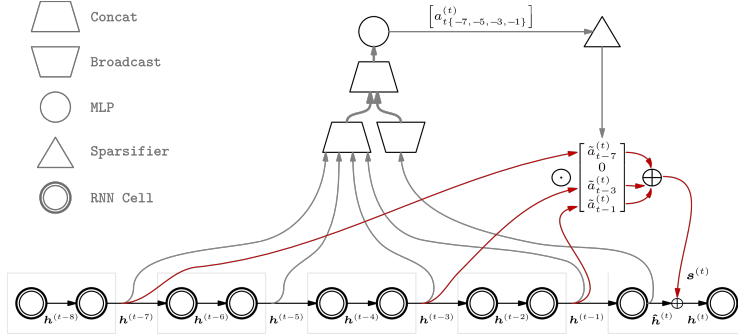

Figure 1: This figure illustrates the forward pass in SAB for the configuration $k_{\text{top}} = 3$, $k_{\text{att}} = 2$, $k_{\text{trunc}} = 2$. This involves sparse retrieval (§ 3.1) and summarization of memories into the next RNN hidden state. Gray arrows depict how attention weights $\boldsymbol{a}^{(t)}$ are evaluated, first by broadcasting and concatenating the current provisional hidden state $\hat{\boldsymbol{h}}^{(t)}$ against the set of all memories $\mathcal{M}$ and computing raw attention weights with an MLP. The sparsifier selects and normalizes only the $k_{\text{top}}$ greatest raw attention weights, while the others are zeroed out. Red arrows show memories corresponding to non-zero sparsified attention weights being weighted, summed, then added into $\hat{\boldsymbol{h}}^{(t)}$ to compute the final hidden state $\boldsymbol{h}^{(t)}$.

# 3 Sparse Attentive Backtracking

Mindful that humans use a very sparse subset of past experiences in credit assignment, and are capable of direct random access to past experiences and their relevance to the present, we present here SAB: the principle of learned, dynamic, sparse access to, and replay of, relevant past states for credit assignment in neural network models, such as RNNs.

In the limit of maximum sparsity (no access to the past), SAB degenerates to the use of a regular static neural network. In the limit of minimum sparsity (full access to the past), SAB degenerates to the use of a full self-attention mechanism. For the purposes of this paper, we explore the gap between these with a specific variety of augmented LSTM models; but SAB does ***not*** refer to any particular architecture, and the augmented LSTM described herein is used purely as a vehicle to explore and validate our hypotheses in §1.

Broadly, an SAB neural network is required to do two things:

- *During the forward pass,* manage a memory unit and select at most a sparse subset of past memories at every timestep. We will call this *sparse retrieval*.

- *During the backward pass,* propagate gradient only to that sparse subset of memory and its local surroundings. We will call this *sparse replay*.

## 3.1 Sparse Retrieval of Memories

Just as humans make a *selective* use of *all* past memories to inform their decisions in the present, so must an SAB model learn to remember and dynamically select only a few memories that could be potentially useful in the present. There are several alternative implementations of this concept. An important class of them are *attention mechanisms*, especially *self-attention* over a model's own past states. Closely linked to the question of dynamic access to memory is the structure of the memory itself; for instance, in the Differentiable Neural Computer (DNC) (Graves et al., 2016), the memory is a fixed-size tensor accessed with explicit read and write operations, while in Bahdanau et al. (2014), the memory is implicitly a list of past hidden states that continuously grows.

For the purposes of this paper, we choose a simple approach similar to Bahdanau et al. (2014). Many other options are possible, and the question of memory representation in humans (faithful to actual brains) and machines (with good computational properties) remains open. Here, to test the principle of SAB without having to answer that question, we use an approach already shown to work well in machine learning. We augment a unidirectional LSTM with the memory of every $k_{att}$'th hidden state from the past, with a modified hard self-attention mechanism limited to selecting at most $k_{top}$ memories at every timestep. Future work should investigate more realistic mechanisms for storing memories, e.g., based on saliency, novelty, etc. But this simple scheme allows us to test the hypothesis that neural network models can still perform well even when compelled at every timestep to access

their past sparsely. If they cannot, then it would be meaningless to further encumber them with a bounded-size memory.

**SAB-augmented LSTM**   We now describe the sparse retrieval mechanism that we have settled on. It determines which memories will be selected on the forward pass of the RNN, and therefore also which memories will receive gradient on the backward pass during training.

At time $t$, the underlying LSTM receives a vector of hidden states $\boldsymbol{h}^{(t-1)}$, a vector of cell states $\boldsymbol{c}^{(t-1)}$, and an input $\boldsymbol{x}^{(t)}$, and computes new cell states $\boldsymbol{c}^{(t)}$ and a *provisional* hidden state vector $\hat{\boldsymbol{h}}^{(t)}$ that also serves as a provisional output. We next use an attention mechanism that is similar to Bahdanau et al. (2014), but modified to produce sparse attention decisions. First, the provisional hidden state vector $\hat{\boldsymbol{h}}^{(t)}$ is concatenated to each memory vector $\boldsymbol{m}^{(i)}$ in the memory $\mathcal{M}$. Then, an MLP with one hidden layer maps each such concatenated vector to a scalar, non-sparse, raw attention weight $a_i^{(t)}$ representing the salience of the memory $i$ at the current time $t$. The MLP is parametrized with weight matrices $\boldsymbol{W}_1$, $\boldsymbol{W}_2$ and $\boldsymbol{W}_3$.

With the raw attention weights, we compute the sparsified attention weights $\tilde{a}_i^{(t)}$ by subtracting out the $(k_{top} + 1)$'th raw weight from all the others, passing the intermediate result through ReLU, then normalizing to sum to 1. This mechanism is differentiable (see S.3 for details) and effectively implements a discrete, hard decision to drop all but $k_{top}$ memories, weigh the selected memories by their *prominence* over the others, as opposed to their raw value. This is different from typical attention mechanisms that normalize attention weights using a softmax function (Bahdanau et al., 2014), whose output is never sparse.

A summary vector $\boldsymbol{s}^{(t)}$ is then computed using a simple sum of the selected memories, weighted by their respective sparsified attention weight. Given that this sum is very sparse, the summary operation is very fast. This summary is then added into the provisional hidden state $\hat{\boldsymbol{h}}^{(t)}$ computed previously to obtain final state $\boldsymbol{h}^{(t)}$.

Lastly, to compute the SAB-augmented LSTM cell's output $\boldsymbol{y}^{(t)}$ at $t$, we concatenate $\boldsymbol{h}^{(t)}$ and summary vector $\boldsymbol{s}^{(t)}$, then apply an affine output transform parametrized with learned weights matrices $\boldsymbol{V}_1$ and $\boldsymbol{V}_2$ and bias vector $\boldsymbol{b}$.

---

**Algorithm 1** SAB-augmented LSTM

1: **procedure** SABCELL $(\boldsymbol{h}^{(t-1)}, \boldsymbol{c}^{(t-1)}, \boldsymbol{x}^{(t)})$
**Require:** $k_{top} > 0,\ k_{att} > 0,\ k_{trunc} > 0$
**Require:** Memories $\boldsymbol{m}^{(i)} \in \mathcal{M}$
**Require:** Previous hidden state $\boldsymbol{h}^{(t-1)}$
**Require:** Previous cell state $\boldsymbol{c}^{(t-1)}$
**Require:** Input $\boldsymbol{x}^{(t)}$
2:    $\hat{\boldsymbol{h}}^{(t)}, \boldsymbol{c}^{(t)} \leftarrow \texttt{LSTMCell}(\boldsymbol{h}^{(t-1)}, \boldsymbol{c}^{(t-1)}, \boldsymbol{x}^{(t)})$
3:    **for all** $i \in 1 \ldots |\mathcal{M}|$ **do**
4:        $\boldsymbol{d}_i^{(t)} \leftarrow \boldsymbol{W}_1 \boldsymbol{m}^{(i)} + \boldsymbol{W}_2 \hat{\boldsymbol{h}}^{(t)}$
5:        $a_i^{(t)} \leftarrow \boldsymbol{W}_3 \tanh(\boldsymbol{d}_i^{(t)})$
6:        $a_{\text{ktop}}^{(t)} \leftarrow \texttt{sorted}(\boldsymbol{a}^{(t)})\,[k_{top}\texttt{+1}]$
7:        $\tilde{\boldsymbol{a}}^{(t)} \leftarrow \text{ReLU}\left(\boldsymbol{a}^{(t)} - a_{\text{ktop}}^{(t)}\right)$
8:        $\boldsymbol{s}^{(t)} \leftarrow \sum\limits_{\boldsymbol{m}^{(i)} \in \mathcal{M}} \tilde{a}_i^{(t)} \boldsymbol{m}^{(i)} \Big/ \sum\limits_i \tilde{a}_i^{(t)}$
9:        $\boldsymbol{h}^{(t)} \leftarrow \hat{\boldsymbol{h}}^{(t)} + \boldsymbol{s}^{(t)}$
10:       $\boldsymbol{y}^{(t)} \leftarrow \boldsymbol{V}_1 \boldsymbol{h}^{(t)} + \boldsymbol{V}_2 \boldsymbol{s}^{(t)} + \boldsymbol{b}$
11:       **if** $t \equiv 0 \pmod{k_{att}}$ **then**
12:           $\mathcal{M}.\texttt{append}(\boldsymbol{h}^{(t)})$
13:       **return** $\boldsymbol{h}^{(t)}, \boldsymbol{c}^{(t)}, \boldsymbol{y}^{(t)}$

---

The forward pass into a hidden state $\boldsymbol{h}^{(t)}$ has two paths contributing to it. One path is the regular sequential forward path in an RNN; the other path is through the dynamic but sparse skip connections in the attention mechanism that connect the present states to potentially very distant past experiences.

## 3.2   Sparse Replay

Humans are trivially capable of assigning credit or blame to events even a long time after the fact, and do not need to replay all events from the present to the credited event sequentially and in reverse to do so. But that is effectively what RNNs trained with full BPTT require, and this does not seem biologically plausible when considering events which are far from each other in time. Even less plausible is TBPTT because it ignores time dependencies beyond the truncation length $k_{trunc}$.

SAB networks' twin paths during the forward pass (sequential connection and sparse skip connections) allow gradient to flow not just from $\boldsymbol{h}^{(t)}$ to $\boldsymbol{h}^{(t-1)}$, but also to the at-most $k_{top}$ memories $\boldsymbol{m}^{(i)}$ retrieved by the attention mechanism (and no others.) Learning to deliver gradient directly (and sparsely) where it is needed (and nowhere else) (1) avoids competition for the limited information-carrying capacity of the sequential path, (2) is a simple form of credit assignment, (3) and imposes

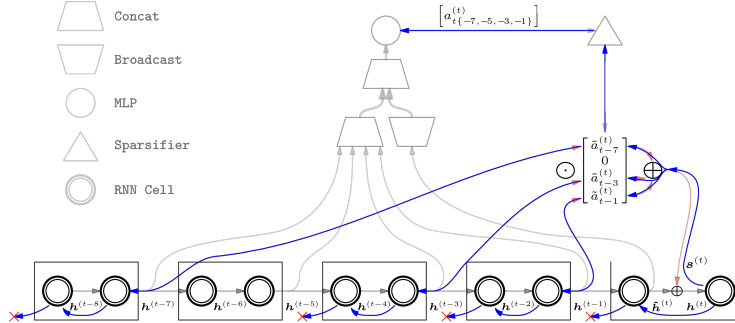

Figure 2: This figure illustrates the backward pass in SAB for the configuration $k_{\text{top}} = 3$, $k_{\text{att}} = 2$, $k_{\text{trunc}} = 2$. The gradients are passed to the hidden states selected in the forward pass and a local truncated backprop is performed around those hidden states. Blue arrows show the gradient flow in the backward pass. Red crosses indicate TBPTT truncation points, where the gradient stops propagating.

a trade-off that is absent in previous, dense self-attentive mechanisms: opening a connection to an interesting or useful timestep must be made at the price of excluding others. This competition for a limited budget of $k_{top}$ connections results in interesting timesteps being given frequent attention and strong gradient flow, while uninteresting timesteps are ignored and starve.

**Mental updates**    If we not only allow gradient to flow directly to a past timestep, but on to a few local timesteps around it as well, we have *mental updates*: a type of local credit assignment around a memory. There are various ways of enabling this. In our SAB-augmented LSTM, we choose to perform TBPTT locally before the selected timesteps ($k_{trunc}$ timesteps before a selected one.)

## 4   Experimental Setup and Results

**Baselines**    We compare SAB to two baseline models for all tasks: 1) an LSTM trained both using full BPTT and TBPTT with various truncation lengths; 2) an LSTM augmented with full self-attention trained using full BPTT. For the pixel-by-pixel Cifar10 classification task, we also compare to the Transformer architecture (Vaswani et al., 2017).

**Copying and adding problems (Q1)**    The copy and adding problems defined in Hochreiter & Schmidhuber (1997) are synthetic tasks specifically designed to evaluate a model's performance on long-term dependencies by testing its ability to remember a sub-sequence for a large number of timesteps.

For the copy task, the network is given a sequence of $T + 20$ inputs consisting of: a) 10 (randomly generated) digits (digits 1 to 8) followed by; b) T blank inputs followed by; c) a special end-of-sequence character followed by; d) 10 additional blank inputs. After the end-of-sequence character the network must output a copy of the initial 10 digits. The adding task requires the model to sum two specific entries in a sequence of $T$ (input) entries. Each example in the task consists of two input vectors of length $T$. The first is a vector of uniformly generated values between 0 and 1. The second vector encodes a binary mask which indicates the two entries in the first input to be added (the mask vector consists of $T - 2$ zeros and 2 ones). The mask is randomly generated with the constraint that masked-in entries must be from different halves of the first input vector.

The hyperparameters for both baselines and SAB are kept the same. All models have 128 hidden units and use the Adam Kingma & Ba (2014) optimizer with a learning rate of 1e-3. The first model in the ablation study (dense version of SAB) was more difficult to train, therefore we explored different learning rates ranging from 1e-3 to 1e-5. We report the best performing model.

The performance of SAB almost matches the performance of LSTMs augmented with self-attention trained using full BPTT. Note that our copy and adding LSTM baselines are more competitive compared to ones reported in the existing literature (Arjovsky et al., 2016). These findings support our hypothesis that at any given time step, only a few past events need to be recalled for the correct prediction of output of the current timestep.

Table 3 reports the cross-entropy (CE) of the model predictions on unseen sequences in the adding task. LSTM with full self-attention trained using BPTT obtains the lowest CE loss, followed by LSTM trained using BPTT. LSTM trained with truncated BPTT performs significantly worse. When $T = 200$, SAB's performance is comparable to the best baseline models. With longer sequences ($T = 400$), SAB outperforms TBPTT, but is outperformed by pure BPTT. For more details regarding the setup, refer to supplementary material.

**Character level Penn TreeBank (PTB) (Q1)**   We follow the setup in Cooijmans et al. (2016) and all of our models use 1000 hidden units and a learning rate of 0.002. We used non-overlapping sequences of 100 in the batches of 32 as in Cooijmans et al. (2016). All models were trained for up to 100 epochs with early stopping based on the validation performance.

We evaluate the performance of our model using the bits-per-character (BPC) metric. As shown in Table 3, SAB's performance is significantly better than TBPTT's and almost matches BPTT, which is roughly what one expects from an approximate-gradient method like SAB.

**Text8 (Q1)**   We follow the setup of Mikolov et al. (2012); we use the first 90M characters for training, the next 5M for validation and the final 5M characters for testing. We train on non-overlapping sequences of length 180. Due to computational constraints, all baselines use 1000 hidden units. We trained all models using a batch size of 64. We trained SAB for a maximum of 30 epochs.

Details about our experimental setup can be found in the supplementary material. Note that we did not carry out any additional hyperparameter search for our model. Table 3 reports the BPC of the model's predictions on the test sets. SAB outperforms LSTM trained using TBPTT. SAB also outperforms LSTM and self-attention trained with TBPTT. For more details, refer to supplementary material.

**Comparison to LSTM + self attention (with truncation)**   While SAB is trained with TBPTT (and the vanilla LSTM+self-attention is not), Here we argue, that training the vanilla LSTM and self attention with truncation works less well on a more challenging Text8 language modelling dataset.

**Permuted pixel-by-pixel MNIST (Q1)**   This task is a sequential version of the MNIST classification dataset. The task involves predicting the label of the image after being given its pixels as a sequence permuted in a fixed, random order. All models use an LSTM with 128 hidden units. The prediction is produced by passing the final hidden state of the network into a softmax. We used a learning rate of 0.001. We trained our model for about 100 epochs, and did early stopping based on the validation set. Our experiment setup can be found in the supplementary material. Table 5 shows that SAB performs well compared to BPTT.

| Method | Test BPC |
|---|---|
| LSTM (full BPTT) | 1.42 |
| LSTM (TBPTT, $k_{trunc}$=5) | 1.56 |
| LSTM (Self Attention with Truncation, $k_{trunc}$=10)) | 1.48 |
| SAB ($k_{trunc}$=10, $k_{top}$=10, $k_{att}$=10) | 1.44 |

Table 1: Bit-per-character (BPC) Results on the test set for Text8 (lower is better).

**CIFAR10 classification (Q1,Q3)**   We test our model's performance on pixel-by-pixel CIFAR10 (no permutation). This task involves predicting the label of the image after being given it as a sequence of pixels. This task is relatively difficult compared to other tasks, as sequences are substantially longer (length 1024.) Our method outperforms Transformers and LSTMs trained with BPTT (Table 5).

**Learning long-term dependencies (Q1)**   Table 2 reports both accuracy and cross-entropy (CE) of the models' predictions on unseen sequences for the copy memory task. The best-performing baseline model is the LSTM with full self-attention trained using BPTT, followed by vanilla LSTMs trained using BPTT. Far behind are LSTMs trained using truncated BPTT. Table 2 demonstrates that SAB is able to learn the task almost perfectly for all copy lengths $T$. Further, SAB outperforms all LSTM baselines and matches the performance of LSTMs with full self-attention trained using BPTT on the copy memory task. This becomes particularly noticeable as the sequence length increases.

**Transfer learning (Q2)**   We examine the generalization ability of SAB compared to full BPTT trained LSTM and LSTM with full self-attention. The experiment is set up as follows: For the copy

| | $k_{\text{trunc}}$ | $k_{\text{top}}$ | Copying (T=100) | | | Copying (T=200) | | | Copying (T=300) | | |
|---|---|---|---|---|---|---|---|---|---|---|---|
| | | | acc. | $CE_{10}$ | CE | acc. | $CE_{10}$ | CE | acc. | $CE_{10}$ | CE |
| **LSTM** | *full BPTT* | | 99.8 | 0.030 | 0.002 | 56.0 | 1.07 | 0.046 | 35.9 | 0.197 | 0.047 |
| | *full self-attn.* | | 100.0 | 0.0008 | 0.0000 | 100.0 | 0.001 | 0.000 | 100.0 | 0.002 | 7.5e-5 |
| | 1 | - | 20.6 | 1.984 | 0.165 | | | | 14.0 | 2.077 | 0.065 |
| | 5 | - | 31.0 | 1.737 | 0.145 | 17.1 | 2.03 | 0.092 | | | |
| | 10 | - | 29.6 | 1.772 | 0.148 | 20.2 | 1.98 | 0.090 | | | |
| | 20 | - | 30.5 | 1.714 | 0.143 | 35.8 | 1.61 | 0.073 | 25.7 | 1.848 | 0.197 |
| | 150 | - | - | - | - | 35.0 | 1.596 | 0.073 | 24.4 | 1.857 | 0.058 |
| **SAB** | 1 | 1 | 57.9 | 1.041 | 0.087 | 39.9 | 1.516 | 0.069 | 43.1 | 0.231 | 0.045 |
| | 1 | 5 | **100.0** | **0.001** | **0.000** | | | | 89.1 | 0.383 | 0.012 |
| | 5 | 5 | **100.0** | **0.000** | **0.000** | **100.0** | **0.000** | **0.000** | **99.9** | **0.007** | **0.001** |
| | 10 | 10 | **100.0** | **0.000** | **0.001** | **100.0** | **0.000** | **0.000** | | | |

Table 2: Test accuracy and cross-entropy (CE) loss performance on the copying task with sequence lengths of T=100, 200, and 300. Accuracies are given in percent for the last 10 characters. $CE_{10}$ corresponds to the CE loss on the last 10 characters. These results are with mental updates; Compare with Table 4 for without.

| **Adding** | | | T=200 | T=400 |
|---|---|---|---|---|
| | $k_{\text{trunc}}$ | $k_{\text{top}}$ | CE | CE |
| **LSTM** | *full BPTT* | | 4.59e-6 | 1.554e-7 |
| | *full self-attn.* | | 5.541e-8 | 4.972e-7 |
| | 20 | - | 1.1e-3 | |
| | 50 | - | 3.0e-4 | |
| | 100 | - | | 6.8e-4 |
| **SAB** | 5 | 5 | 4.26e-5 | |
| | 5 | 10 | | 2.30e-4 |
| | 10 | 10 | **2.0e-6** | 1.001e-5 |

| **Language** | | | | PTB | Text8 |
|---|---|---|---|---|---|
| | $k_{\text{trunc}}$ | $k_{\text{top}}$ | $k_{\text{att}}$ | BPC | BPC |
| **LSTM** | *full BPTT* | | | 1.36 | 1.42 |
| | 1 | - | - | 1.47 | |
| | 5 | - | - | 1.44 | 1.56 |
| | 20 | - | - | 1.40 | |
| **SAB** | 10 | 5 | 10 | 1.42 | 1.47 |
| | 10 | 10 | 10 | 1.40 | 1.45 |
| | 20 | 5 | 20 | 1.39 | 1.45 |
| | 20 | 10 | 20 | 1.37 | 1.44 |

Table 3: Performance on the adding task (left) and language modeling tasks (PTB and Text8; right). The adding task performance is evaluated on unseen sequences of the T = 200 and T = 400 (note that all methods have configurations that allow them to perform near optimally.) For T = 400, BPTT slightly outperforms SAB, which outperforms TBPTT. For the language modeling tasks, the BPC score is evaluated on the test sets of the character-level PTB and Text8.

task of length $T = 100$, we train SAB, LSTM trained with BPTT, LSTM and full self-attention to convergence. We then take the trained model and evaluate them on the copy task for an array of larger $T$ values. The results are shown in Table 6. Although all 3 models have similar performance on $T = 100$, it is clear that performance for all 3 models drops as $T$ grows. However, SAB still manages to complete the task at $T = 5000$, whereas by $T = 2000$ both vanilla LSTM and LSTM with full self-attention do no better than random guessing ($1/8 = 12.5\%$).

**Importance of sparisity and mental updates (Q4)** We study the necessity of sparsity and *mental updates* by running an ablation study on the copying problem. The ablation study focuses on two variants. The first model attends to all events in the past while performing a truncated update. This can be seen either as a dense version of SAB or an LSTM with full self-attention trained using TBPTT. Empirically, we find that such models are both more difficult to train and do not reach the same performance as SAB. The second ablation experiment tests the necessity of mental updates, without which the model would only attend to the past time steps without passing gradients through them to preceding time steps. We observe a degradation of model performance when blocking gradients to past events. This effect is most evident when attending to only one timestep in the past ($k_{top} = 1$).

We evaluate SAB on language modeling, with the Penn TreeBank (PTB) (Marcus et al., 1993) and Text8 Mahoney (2011) datasets. For models trained using truncated BPTT, the performance drops as $k_{\text{trunc}}$ shrinks. We found that on PTB, SAB with $k_{\text{trunc}} = 20$, $k_{\text{top}} = 10$ performs almost as well as full BPTT. For the larger Text8 dataset, SAB with $k_{\text{trunc}} = 10$ and $k_{\text{top}} = 5$ outperforms LSTM trained using BPTT.

| Ablation | | Copying, T=100 | | | Adding, T=200 CE |
|---|---|---|---|---|---|
| $k_{\text{trunc}}$ | $k_{\text{top}}$ | acc. | $CE_{\text{last 10}}$ | CE | |
| **no MU** 1 | 1 | 49.0 | 1.252 | 0.104 | |
| 5 | 5 | 98.3 | 0.042 | 0.0036 | |
| 10 | 10 | 99.6 | 0.022 | 0.0018 | 2.171e-6 |
| 5 | all | 40.5 | 1.529 | 0.127 | |

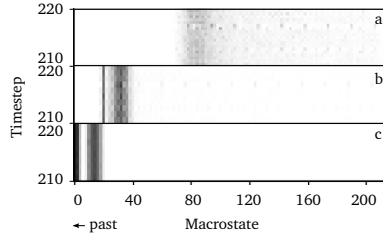

Table 4: Left: ablation studies on the adding and copying tasks. The limiting cases of dense attention ($k_{\text{top}} =$ all) and of no mental updates (**MU**) were tested. Right: focus of the attention for the T=200 copying task, where reproduction of the inital 10 input symbols is required (black corresponds to stronger attention weights). The was generated at different points in training (a-c) within the first epoch. Attention quickly shifts to the relevant parts of the sequence (the initial 10 states.)

| Image class. | | | pMNIST | CIFAR10 |
|---|---|---|---|---|
| $k_{\text{trunc}}$ | $k_{\text{top}}$ | $k_{\text{att}}$ | acc. | acc. |
| **LSTM** *full BPTT* | | | 90.3 | 58.3 |
| 300 | - | - | | 51.3 |
| **SAB** 20 | 5 | 20 | 89.8 | |
| 20 | 10 | 20 | 90.9 | |
| 50 | 10 | 50 | **94.2** | |
| 16 | 10 | 16 | | **64.5** |
| Transformer (Vasvani'17) | | | **97.9** | 62.2 |

Table 5: Test accuracy for the permuted MNIST and CIFAR10 classification tasks.

**Transfer Learning Results**

| Copy len. (T) | LSTM | LSTM +self-a. | SAB |
|---|---|---|---|
| 100 | 99% | 100% | 99% |
| 200 | 34% | 52% | **95%** |
| 300 | 25% | 28% | **83%** |
| 400 | 21% | 20% | **75%** |
| 2000 | 12% | 12% | **47%** |
| 5000 | 12% | OOM | **41%** |

Table 6: Transfer performance (Accuracy for last 10 digits) for models trained on $T = 100$ copy memory task. Comparisons to LSTM and LSTM with full self-attention trained with BPTT.

**Comparison to Transformer (Q3)** We test how SAB compares to the Transformer model (Vaswani et al., 2017), based a self-attention mechanism. On pMNIST, the Transformer model outperforms our best model, as shown in Table 5. On CIFAR10, however, our proposed model performs much better.

## 5 Conclusions

By considering how brains could perform long-term temporal credit assignment, we developed an alternative to the traditional method of training recurrent neural networks by unfolding of the computational graph and BPTT. We explored the hypothesis that a reminding process which uses the current state to evoke a relevant state arbitrarily far back in the past could be used to effectively teleport credit backwards in time to the computations performed to obtain the past state. To test this idea, we developed a novel temporal architecture and credit assignment mechanism called SAB for Sparse Attentive Backtracking, which aims to combine the strengths of full backpropagation through time and truncated backpropagation through time. It does so by backpropagating gradients only through paths for which the current state and a past state are associated. This allows the RNN to learn long-term dependencies, as with full backpropagation through time, while still allowing it to only backtrack for a few steps, as with truncated backpropagation through time, thus making it possible to update weights as frequently as needed rather than having to wait for the end of very long sequences.

Cognitive processes in reminding serve not only as the inspiration for SAB, but suggest two interesting directions of future research. First, we assumed a simple content-independent rule for selecting hidden states for inclusion in the memory (select at every $k_{\text{att}}$ step), whereas humans show a systematic dependence on content: salient, extreme, unusual, and unexpected experiences are more likely to be stored and subsequently remembered. These landmarks of memory should be useful for connecting past to current context, just as an individual learns to map out a city via distinctive geographic landmarks. Second, SAB determines the relevance of past hidden states to the current state through a generic, flexible mapping, whereas humans perform similarity-based retrieval. We conjecture that a version of SAB with a strong inductive bias in the mechanism to select past states may further improve its performance.

**Acknowledgements**

The authors would like to thank Hugo Larochelle, Walter Senn, Alex Lamb, Remi Le Priol, Matthieu Courbariaux, Gaetan Marceau Caron, Sandeep Subramanian for the useful discussions, as well as NSERC, CIFAR, Google, Samsung, SNSF, Nuance, IBM, Canada Research Chairs, National Science Foundation awards EHR-1631428 and SES-1461535 for funding. We would also like to thank Compute Canada and NVIDIA for computing resources. The authors would also like to thank Alex Lamb for code review. The authors would also like to express debt of gratitude towards those who contributed to Theano over the years (now that it is being sunset), for making it such a great tool.

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
