[Supplementary Material · SAB_supplementary.pdf]

# S  Supplementary Material

## S.1  Computational Complexity of *SAB*

If the memory was allowed to grow unbounded in size, then the computational complexity would scale linearly with the length of history. However, humans have a bounded memory. In a computer science context with unbounded memory, the time complexity of the forward pass of both training and inference in SAB is $O(t^2 n^2)$, with $t$ the number of timesteps and $n$ the size of the hidden state. The space complexity of the forward pass of training is unchanged at $O(tn)$, but the space complexity of inference in SAB is now $O(tn)$ rather than $O(n)$. However, the time cost of the backward pass of training cost is very difficult to formulate. Hidden states depend on a sparse subset of past microstates, but each of those past microstates may itself depend on several other, even earlier microstates. The web of active connections is, therefore, akin to a directed acyclic graph, and it is quite possible in the worst case for a backpropagation starting at the last hidden state to touch all past microstates several times. However, if the number of microstates truly relevant to a task is low, the attention mechanism will repeatedly focus on them to the exclusion of all others, and pathological runtimes will not be encountered.

## S.2  Gradient Flow

Our method approximates the true gradient but in a sense it's no different than the kind of approximation made with truncated gradient, except that instead of truncating to the last $k_{trunc}$ time steps, we truncate to one skip-step in the past, which can be arbitrarily far in the past. This provides a way of combating exploding and vanishing gradient problems by learning long-term dependencies. To verify the fact, we ran our model on all the datasets (Text8, Pixel-By-Pixel MNIST, char level PTB) with and without gradient clipping. We empirically found, that we need to use gradient clipping only for text8 dataset, for all the other datasets we observed little or no difference with gradient clipping.

## S.3  Differentiability of the Attention Mechanism

Under Sparse Attentive Backtracking, there is some nuance in the differentiability of a memory access. Specifically,

- The *computation of the memory attention weights* is differentiable.
- The *selection* of the $k_{top}$ most salient memories is *hard, non-differentiable*. In particular, because the selection resembles a $k_{top}$-argmax over $T - 1$ discrete categories, there exists no error signal that rewards or penalizes an SAB network for selecting the timestep at offset $\mathrm{d}t$ from the chosen read-addresses.
- The *selected memories* are nevertheless differentiable, because gradient flows backwards from the memory mechanism's output (the uses of a memory) to its input (the creation of a memory).
- While the selection process for past memories at every timestep is non-differentiable, the past is nevertheless still explored effectively, even if stochastically. During early training, the memory selection mechanism exhibits no strong preferences, with uniformly-random and poor-quality choices. As training progresses, the network encounters useful, relevant memories, and the differentiable attention-weight computation learns to focus on them.