[Reviews · NeurIPS 2018]

Reviewer 1



The authors augment an RNN with skip connections in time, that are sparsely gated by learnable attention. This allows to reap the benefits of full BPTT while effectively using only truncated BPTT. While other forms of attention-gated skip connections in time have been suggested before, to which the authors compare, here the authors looked at sparse (still differentiable) retrieval where only a few top memory entries are selected, enabling the benefits of backpropagating over only a few selected earlier states. Overall, I think this work is very significant, both for enabling faster implementations of BPTT when considering long time horizons, but also for suggesting future directions for how the brain might perform credit assignment and for pointing out further brain strategies / biases to employ in machine learning. With some clarifications / changes as below, I recommend the acceptance of this article for NIPS. 1. In lines 58-60, the authors say that BPTT would require "playing back these events". However, it is not so simple. The error must be backpropagated using the same recurrent weights, and combined with the activities and derivatives at previous time points to modify the weights. The authors should highlight this, else BPTT in the brain seems a simple matter of replaying past memories which is still bio-plausible, even though requires long time scales, whereas BPTT is far less bio-plausible, from perspectives of architecture and electro-physiology. 2. In lines 176 to 181, the authors show how a hard attention mechanism is implemented retrieving only the k_{top} memories. The authors should clarify how this memory access is differentiable, as the usage of soft-attention by other memory-augmented implementations was partly to enable differentiable access allowing backprop to pass through, and would allow the learning of what to attend to in the past given current hidden state \hat{h}. It is not clear here how the authors maintain differentiability while keeping hard attention. Minor: l 156: "computationsl" l 223" "pixel-by-pixle"

Reviewer 2



This work deals with a memory system that is intended to be able to use relevant pieces of information from the potentially distant past in order to inform current predictions. The work is well motivated by examples of biological systems, but even without these arguments, memory systems are of course critical parts of many machine learning systems so any significant contributions here are relevant to almost the entire community. I find the architecture appealing and I think it has features that intuitively should help with the type of settings they consider, which are difficult for standard RNNs due to the time horizons which are found in training in practice. This is also borne out in the empirical results they show in a number of very different domains. On the other hand, I do have some reservations, which lie with the comparisons made both in the Related Work / section 2, and Experimental Results / Section 4. Section 2 only mentions resenets and dense networks (originally for convnets, although the connection to RNNs here is clear), and transformer networks which replace recurrence with attention. I do feel that this section almost completely ignores innovations in memory systems that do allow information to flow more easily over long periods such as hierarchical/multiscale RNNs, among others. Likewise these comparsions are also missing from the experimental results; outperforming an LSTM is not so impressive when other architectures with similar design considerations exist. -- After rebuttal -- I appreciate your comments and work responding to my, and other reviewers queries. The additional work is significant both in terms of effort and quality of results and warrants a revision of score.

Reviewer 3



In this paper, the authors present a novel framework, Sparse Attentive Backtracking (SAB), which can be used to capture long term dependencies in sequential data as an alternative to Back Propagation Through Time (BPTT). The inspiration for the framework comes from the intuition of how humans do credit assignment — the relevant memories are retrieved based on the current stimulus and accredited based on their association with the stimulus, which is quite different from BPTT which rolls out the past states in reverse order. The paper describes the biological motivation and details how certain memories (past hidden states) can be aggregated into a summary with weights based on their association with the current state. The paper is clearly written and the workings of the framework aptly exposed. There is a slight disconnect between the motivation which talks about attending to salient memories within a long horizon, and the proposal which limits to hand coded connections to a set of memories. However, even with this limitation, an LSTM augmented with SAB performs comparably to state of the art methods often outperforming them which hints at its utility and the potential for improved results with better curated memories. Furthermore, the authors talk about these limitations in venues for future work alleviating my main concern to some extent. Overall, I think the paper makes a novel contribution which would be of interest to the wider community even though some more work on addressing the shortcomings of the presented approach would make for an even stronger submission. -- After rebuttal -- Thanks for addressing the different concerns raised by the reviewers, and doing the extra work to argue the relevance of the proposed method. I maintain that this is a good paper which would be interesting to the scientific community.